# Evaluation of Dram Score as a Predictor of Poor Postoperative Outcome in Spine Surgery

**DOI:** 10.3390/jcm9123825

**Published:** 2020-11-26

**Authors:** Antonio Serrano-García, Manuel Fernández-González, Jesús Betegón-Nicolás, Julio Villar-Pérez, Ana Lozano-Muñoz, José Hernández-Encinas, Ignacio Fernández-Bances, Marta Esteban-Blanco, Jesús Ángel Seco-Calvo

**Affiliations:** 1Psychiatry Service, Department of Psychosomatic, Complejo Asistencial Universitario de León, 24008 León, Spain; 2Spine Unit, Department of Orthopedic Surgery and Traumatology, Complejo Asistencial Universitario de León, 24008 León, Spain; mfdezg1990@gmail.com (M.F.-G.); jbetegonnicolas@gmail.com (J.B.-N.); javillarp@gmail.com (J.V.-P.); analozanomunoz@yahoo.es (A.L.-M.); jahencinas@yahoo.es (J.H.-E.); docbances@gmail.com (I.F.-B.); marta270184@gmail.com (M.E.-B.); 3Institute of Biomedicine (IBIOMED), University of León, Campus de Vegazana, 24071 Leon, Spain; jesus.seco@unileon.es; 4Department of Physiology, Visiting Researcher of Basque Country University, 48940 Leioa, Spain

**Keywords:** scoring DRAM, back pain, radicular pain, physical health, mental health, psychosocial factors

## Abstract

The Distress Risk Assessment Method (DRAM) was presented by Main, Wood and Hillis in 1992 as a simple means of assessing the risk of failure due to psychosocial factors in spine surgery. To our knowledge, it has not been used in our setting. The aim of this study was to analyse the usefulness of the Spanish translation of this instrument to predict poor outcomes. Methods: A prospective blind study was conducted including 65 patients undergoing spine surgery. We created two groups of patients based on DRAM score: not distressed (NDRAM) or distressed (DDRAM). A visual analogue scale for pain and the 12-Item Short Form Health Survey (SF-12) were used at baseline, 6 weeks and 6 months. Results: 24 patients were classified as DDRAM and 38 as NDRAM, with 3 patients not completing the questionnaires. The analysis found no significant differences in the demographic or clinical variables at baseline. At 6 weeks and 6 months, the NDRAM group showed improvements in low back pain (*p* < 0.001; *p* = 0.005), leg pain (*p* < 0.001; *p* = 0.017), physical health (*p* = 0.031; *p* = 0.003) and mental health (*p* = 0.137; *p* = 0.049). In contrast, in the DDRAM group, though leg pain score improved (*p* < 0.001; *p* = 0.002), there was no improvement at 6 weeks or 6 months in low back pain (*p* = 0.108; *p* = 0.287), physical health (*p* = 0.620; *p* = 0.263) or mental health (*p* = 0.185; *p* = 0.329). Conclusions: In our setting, the DRAM is a useful screening tool, and it has allowed the creation of a program between psychiatry and spine surgery.

## 1. Introduction

Back pain is a highly prevalent condition, with estimates suggesting that at least 84% of people suffer from this type of pain at some point in their life [1]. There is a paucity of data on the prevalence of spinal deformity, but Schwab et al. indicated that the prevalence of adult spinal deformity is between 2 and 32% of the population [2].

It has long been known that patients with psychological comorbidity often have disability related to chronic back pain [3,4,5]. Several studies have indicated that patients with psychosocial comorbidities, including depression, somatization and anxiety [6], have worse outcomes after spine surgery, including spinal fusion [7,8,9], laminectomy [10] and lumbar discectomy [4,5,11,12]. Walker analysed the role of depression at 12 months after spinal disc surgery and concluded that patients with fewer depressive symptoms had significantly less clinical pain than those with more depressive symptoms [1]. Similarly, another study found that patients with fewer depressive symptoms and less work stress were more likely to return to work within 2 years after laminectomy surgery than patients who were more depressed [13].

Recent years have seen a growing interest in measuring the perspective of patients in the evaluation of disease and the benefits of treatment. In particular, measures of health-related quality of life are becoming recognised as key tools for assessing the safety and effectiveness of surgery [14,15]. The Medical Outcomes Study 36-Item Short Form Health Survey SF-36 has become the most widely used generic measure [16]. Although the completion time for the SF-36 is only 5 to 10 min, it may place an unreasonable burden on respondents when applied together with other instruments or in situations with limitations on time. The two summary components of the SF-36 were used to develop a shorter version of the questionnaire, the SF-12, which can be answered in approximately 2 minutes [17].

The Distress Risk Assessment Method (DRAM) was proposed by Main et al. in 1992 [18] and it is composed of a combination of two self-administered scales, the Modified Zung Depression Index [19] and the Modified Somatic Perception Questionnaire (MSPQ) [20]. The utility of the DRAM has been tested in numerous clinical trials and the method has been shown to be effective for predicting the risk of poor postoperative outcomes in various spine procedures [21,22,23]. Specifically, Chaichana et al. [21] found that clinically depressed patients (Zung score > 33, on the American version of the scale) were significantly less likely to experience clinically relevant improvement (*p* < 0.05) in leg and back pain and in disability and quality of life. In line with this, patients with higher somatization scores (MSPQ > 12) were significantly less likely to experience clinically relevant improvement (*p* < 0.05) in leg and back pain, as well as disability and quality of life.

The study of Licciardone et al. [23] indicated that spine surgeons are not well trained in detecting psychosocial problems. In this context, we consider it is necessary to explore tools to improve the detection of this kind of problem. Given all this, the aim of this study was to analyse the effectiveness of the Spanish version of the DRAM for assessing psychosocial factors as a predictor of poor clinical outcomes, which would be useful in routine clinical practice.

## 2. Material and Methods

This prospective blinded study was approved by the Ethics Committee for Clinical Research of León (11 July 2016 Reference: 1668). The study population was composed of consecutive patients on the waiting list for elective spine surgery in the Spine Unit at the Hospital of León between August 2016 and February 2017. To be eligible for inclusion, patients had to be at least 18 years old. Exclusion criteria were inability to understand or complete the questionnaires by themselves. All patients were informed about the study and provided written informed consent. We have selected DRAM among the available tools because it was specifically designed for spine surgery, is self-applied and has extensive experience of use in other countries. The surgeons that indicated the procedure remained blind to DRAM scores throughout the study and no changes were done in the usual clinical intervention. Patients were assessed at baseline, 6 weeks and 6 months after surgery using a visual analogue scale (VAS) for pain and the SF-12.

Two groups of patients were created based on DRAM score: Group 1: Not distressed (NDRAM, including the classifications normal and at-risk) and Group 2: distressed (DDRAM, including the classifications depressive and somatic) (Table 1).

In addition to the DRAM score, we assessed the presence of other clinical conditions including adjacent vertebral fracture, adjacent segment disease, implant rupture, pseudarthrosis and other complications (including infections and screw malposition).

Postoperative results were considered poor if patients met any of the following criteria in the first year after surgery: further surgery as a consequence of a complication, referral for new surgery, death for any cause or a new admission for any reason in the Spine Unit, and/or hospital stay of longer than 20 days in case of fusion of three levels or more and 10 days in cases of fusion of two levels.

Further, the following were recorded: any comorbidities, including anxiety or depressive disorders, asthma, autoimmune diseases, diabetes, fibromyalgia, hypercholesterolemia, hypertriglyceridemia, hypertension, kidney failure, osteoporosis, severe mental disorders and thyroid disease, and whether patients were a smoker or problem drinker. All this information was obtained from medical records. 

An Excel^®^ for Windows database was created for the study and the statistical analysis was performed with IBM SPSS Statistics for Windows, version 23 (IBM Corp., Armonk, NY, USA). In order to know if both groups had a normal distribution of main study variables (back pain Visual Analogic Scale (VAS) score, leg pain VAS score, SF-12 physical health score and SF-12 mental health score) at baseline, we performed a Kolmogorov-Smirnov test. We also performed two paired Student’s *t*-tests (baseline–six weeks and baseline–six months) to compare result variables in the follow-up. We also compared also categorical sociodemographic and clinical variables at baseline in order to know if exists any condition that is differently distributed between the two groups. *p* values < 0.05 were considered significant. 

## 3. Results

Initially, 65 patients were included in the study, but 3 did not complete the questionnaires correctly. Finally, 38 patients (58.5%) were classified into the NDRAM group and 24 (36.9%) into the DDRAM group. Baseline characteristics were similar in the two groups. Of the 65 patients, 33 were men (50.77%) and 32 were women (49.23%). They had a mean age of 55.56 years (range: 32–80 years), mean height of 1.64 meters (±0.09) and mean weight of 75.91 kilograms (±13.51). Table 2 shows demographic and clinical characteristics of patients grouped by DRAM score. We found no significant differences between NDRAM and DDRAM groups in the percentage of patients who had a history of anxiety or depressive disorder (chi^2^
*p* = 0.059), asthma (chi^2^
*p* = 0.205), diabetes (chi^2^
*p* = 0.776), hypercholesterolemia (chi^2^
*p* = 0.632), hypertriglyceridemia (chi^2^
*p* = 0.815), hypertension (chi^2^
*p* = 0.679), kidney failure (chi^2^
*p* = 1), osteoporosis (chi^2^
*p* = 1), severe mental disorders (chi^2^
*p* = 0.702), spine surgery (chi^2^
*p* = 0.717), or thyroid disease (chi^2^
*p* = 0.059), or who were a current smoker (chi^2^
*p* = 0.535), ex-smoker (chi^2^
*p* = 0.958), or problem drinker (chi^2^
*p* = 0.205). We did, however, detect significant differences between the two groups in the percentage of patients with a history of autoimmune diseases (chi^2^
*p* = 0.025) and fibromyalgia (chi^2^
*p* = 0.025), in both cases the percentage being higher in the DDRAM group.

At baseline, the main study variables, namely, the low back pain VAS score and SF-12 Physical Health Composite score (PCS) (mean ± standard deviation) were similar in the two groups (low back pain score = 7.87 (±1.88) and SF-12 PCS = 24.11 (±4.91) in the NDRAM group vs low back pain score = 8.00 (±2.04) and SF-12 PCS = 25.37 (±4.28) in the DDRAM group; *p* = 0.796 and *p* = 0.311 respectively). On the other hand, the leg pain VAS score and SF-12 Mental Health Composite score (MCS) differed significantly between the groups (leg pain score = 7.53 (±2.09) and SF-12 MCS = 38.43 (± 10.87) in the NDRAM group vs. leg pain score = 8.54 (±1.35) and SF-12 MCS = 27.63 (±9.71) in the DDRAM group; *p* = 0.039 and *p* < 0.001, respectively).

In follow-up, the mean low back pain scores in the DDRAM group had not significantly changed at 6 weeks (*p* = 0.108) or 6 months (*p* = 0.287), but in the NDRAM group, we found significant changes at both 6 weeks (*p* < 0.001) and 6 months (*p* = 0.005) (Figure 1). In the case of leg pain, patients in both groups showed significant improvements at 6 weeks and 6 months (respectively *p* < 0.001 and *p* = 0.002 for the DDRAM group and *p* < 0.001 and *p* = 0.017 for the NDRAM group) (Figure 2).

Analysing the SF-12 PCS (Figure 3), no significant changes were observed in the DDRAM group at 6 weeks (*p* = 0.620) or 6 months (*p* = 0.263), while the NDRAM group showed significant improvements at 6 weeks (*p* = 0.031) and 6 months (*p* = 0.003). Considering the SF-12 MCS (Figure 4), again, no significant changes were observed in the DDRAM group at 6 weeks (*p* = 0.185) or 6 months (*p* = 0.329), and scores had not improved significantly in the NDRAM group by 6 weeks (*p* = 0.137) but the improvement was significant at 6 months (*p* = 0.003) (Figure 4).

The rates of clinical complications did not differ significantly between the groups (chi^2^
*p* = 0.120). Only one patient underwent reoperation; the procedure was indicated for infection, which resolved after this treatment.

## 4. Discussion

This study provides the first evidence for a potential role of the DRAM in routine clinical practice in our setting. Specifically, the DRAM score has a high predictive value in spine surgery, successfully identifying patients who may have poor surgical outcomes. This method was selected because there is extensive evidence from its use in other countries, it is simple to use and it is able to provide information on two clinical conditions that may significantly influence the indication and the success of a surgical procedure, namely, depression and somatization [22]. Many other studies have assessed the relationship between depression and poor surgical outcomes [4,7,8] but we have found no research on the systematic use of the MSPQ and Zung scores to evaluate psychosocial issues in a Spanish population. Based on our results, we can state that patients in our setting classified as distressed using the DRAM before surgery have poorer health-related quality of life or do not experience improvements that might be expected after surgery.

Among the main limitations of our study, it should be noted that some patients did not complete the DRAM scoring correctly and that an association was found between DDRAM and the presence of autoimmune diseases, which could indicate that there were false positives among patients affected by these conditions. There is also a certain resistance to go to psychiatric services on the part of the population. As DRAM scoring is a self-applied tool, it allows the spine surgeon to have an initial psychological evaluation without the intervention of a psychiatrist, which gives the surgeon this information without having to overcome the patient’s resistance.

One important difference was observed between the two types of pain parameters measured. In the DDRAM group, no improvements were observed in back pain while radicular pain (leg pain) did improve, although baseline scores indicated that they had significantly greater perception of pain. In our view, back pain is a clinically more diffuse condition, which is influenced to varying extent by a wide range of factors, that is, such pain may derive from a vertebral fracture but also from muscle contracture, osteoarthritis, anxiety, etc. In contrast, radicular pain is almost exclusively attributable to stenosis causing compression of a nerve root. At baseline, the DDRAM group reported more pain and this could be related to a perceived increase in unpleasant sensations related to changes in mood.

Regarding health-related quality of life, the NDRAM group showed considerable improvements, including in mental health, that could be attributed to the impact of somatic disturbance on mood and overall functioning of the patient, this being consistent with the late improvement observed. Patients must readapt themselves to improved performance, something which might not occur on a timescale of 6 weeks but be observable by 6 months. Nonetheless, the DDRAM group showed no improvement in quality-of-life parameters, with physical health or mental health, at any time point. Given this, we believe that spine surgery should not be indicated in this group of patients if the goal is to improve these parameters.

The percentage of patients classified as distressed using the DRAM was notably high. In view of the results, we consider that it is essential to adopt a multidisciplinary approach to spine problems, involving collaboration between different specialists, because the presence of a somatic condition does not exclude psychiatric problems and vice versa. In order to indicate surgical procedures, we must take psychosocial factors into account because they may cause the procedure to fail [24]. The DRAM score can be used as a simple screening test that provides spine specialists with complementary information about factors that influence the outcome of their patients. Today, it is inconceivable to indicate a surgical procedure without an imaging test; perhaps, it should also be inconceivable without psychosocial screening.

In our study, 38% of patients in whom surgery was indicated were classified as distressed using the DRAM and did not experience significant improvements in back pain postoperatively. At baseline, leg pain scores in distressed patients were significantly higher than in non-distressed patients, but both groups reported that leg pain improved after surgery. Neither physical nor mental health improved over the follow-up in patients classified as distressed, while both improved in other patients, although improvements in mental health appeared later. In our setting, the use of the DRAM scoring has allowed the creation of a collaborative program between the psychosomatic and spinal surgery services (Figure 5). The DRAM tool works as a screening that surgeons apply to any patient with a potential surgical indication, from then on all cases are discussed in a multidisciplinary clinical session in which the entire spinal surgery team and a psychiatrist trained in pain management and psychosomatic medicine are present. Once the case has been discussed, two options can be taken, surgical intervention or psychosomatic consultation. Once the patient has been evaluated in the psychosomatic psychiatry consultation, the patient can be evaluated again in the multidisciplinary clinical session if his psychopathological situation changes. It is essential not to consider the psychiatric intervention as a support, psychiatric couselling and interventions should be integral part of the of the low back pain treatment.

## Figures and Tables

**Figure 1 jcm-09-03825-f001:**
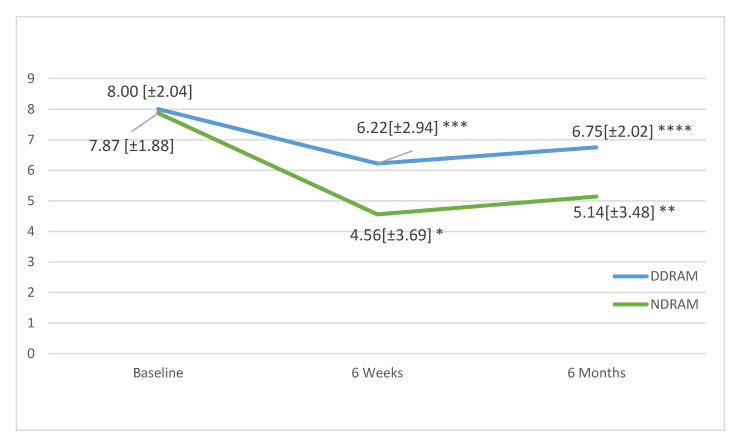
Change in low back pain as measured on a visual analogue scale in each study group throughout the follow-up. Scores expressed as means and standard deviations. *p*-values calculated with paired Student’s *t*-test. * *p* < 0.001, ** *p* < 0.005, *** Not Significant, **** Not Significant.

**Figure 2 jcm-09-03825-f002:**
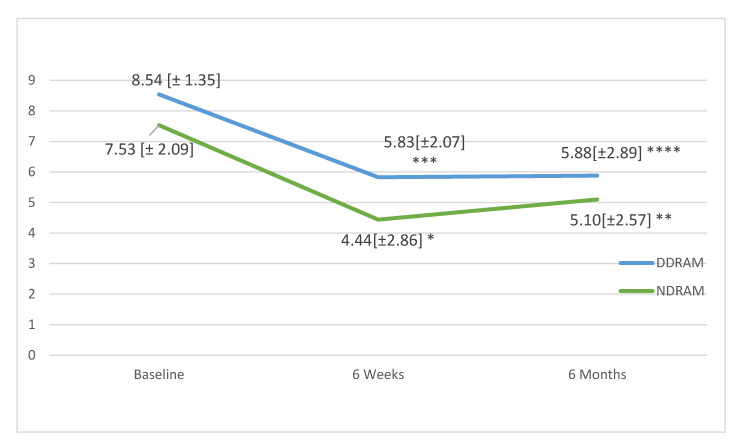
Change in low back pain as measured on a visual analogue scale in each study group throughout the follow-up. Values expressed in means and standard deviations. *p*-values calculated with paired Student’s *t*-test. * *p* <0.001, ** *p* = 0.017, *** *p* < 0.001, **** *p* = 0.002.

**Figure 3 jcm-09-03825-f003:**
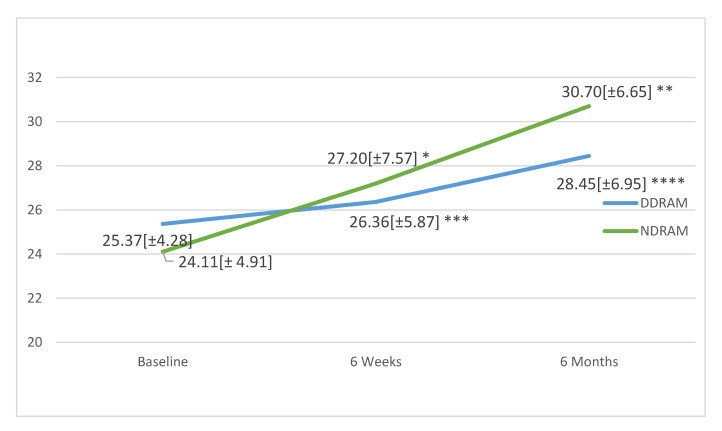
Change in physical health as measured with the Short Form 12 Health Survey (SF-12) in patients in each study group throughout the follow-up. Values expressed in means and standard deviations. *p*-values calculated with paired Student’s *t*-test. * *p* = 0.031, ** *p* = 0.003, *** *p* = 0.620, **** *p* = 0.263.

**Figure 4 jcm-09-03825-f004:**
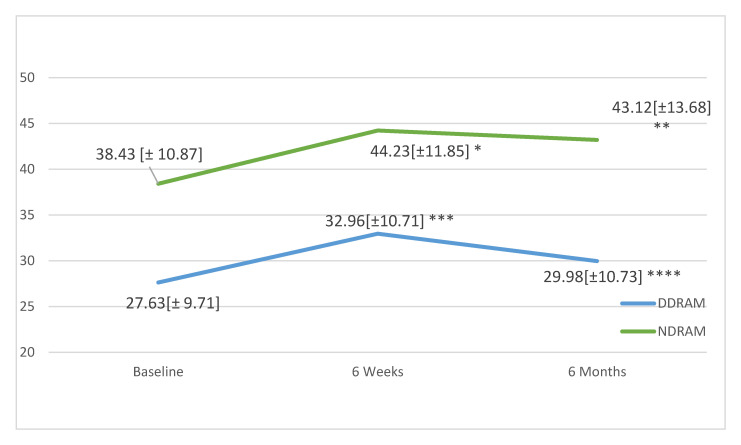
Changes in mental health as measured with the SF-12 in patients in each study group throughout the follow-up. Values expressed in means and standard deviations. *p*-values calculated with paired Student’s *t*-test. * *p* = 0.137, ** *p* = 0.049, *** *p* = 0.185, **** *p* = 0.329.

**Figure 5 jcm-09-03825-f005:**
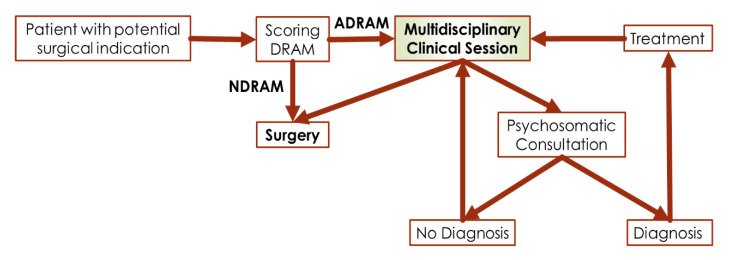
Structure of the multidisciplinary clinical management of low back pain in our setting.

**Table 1 jcm-09-03825-t001:** The Distress Risk Assessment Method (DRAM): classification of patients according to the scores obtained on the Modified Somatic Perception Questionnaire (MSPQ) and Zung Depression index.

Classification	Cut-Offs
Normal	Zung score < 36
At risk	Zung score 36–51 MSPQ score < 12
Distressed: depressive	Zung score > 52
Distressed: somatic	Zung score 36–51 and MSPQ score ≥ 12

**Table 2 jcm-09-03825-t002:** Baseline demographic and clinical characteristics of patients included in the study stratified by DRAM score.

	N = 65		
	DDRAM Group *n* = 24	NDRAM Group *n* = 38	*p*-Value
Age, years	56.96 (±13.89)	54.70 (±12.53)	0.513
Sex			
Men, %	13.85% (9)	33.85% (22)	0.118
Women, %	23.08 % (15)	24.62% (16)	
Surgical time (minutes)	163.96 (±60.85)	162.50 (±83.31)	0.941
Estimated blood loss (cc)	480.43 (±234.39)	436.46 (±228.02)	0.475
Hospital stay (days)	7.92 (±2.62)	8.29 (±4.87)	0.732
Number of instrumented levels	1.71(±1.074)	1.97(±1.68)	0.492

DRAM: Distress and Risk Assessment Method; DDRAM: distressed based on DRAM score; NDRAM: not distressed based on DRAM score.

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
