# Peer review of "Evaluation of Dram Score as a Predictor of Poor Postoperative Outcome in Spine Surgery"

_jcm, 2020, doi:10.3390/jcm9123825_

Round 1

Reviewer 1 Report

I think that this study open an important point of view regarding with the pre-operative  correct selection of the Patients.

Anyway, according my humble opinion,

I have two questions.

1) I think that more informations must be given about the characteristics of the two groups. Are they sufficiently homogeneous? Otherwise we are not sure that they must be compared.

2) the second observation have an ethical features. When you identify a patients that, according to this classification, is “not suitable” for surgery ( from a psychological point of view), how  should you use this information? Please, provide some clinical practice use of this informations.

Thank you.

Author Response

To the reviewer number 1:

Thanks you for your comments, I will try to explain the modifications I have done and to answer to your questions:

We have expanded the description of statistical methods to improve the comprehensibility of the operations performed.

  • I think that more informations must be given about the characteristics of the two groups. Are they sufficiently homogeneous? Otherwise we are not sure that they must be compared.

The DRAM score automatically classified patients into one of the two comparison groups. From this classification the statistical operations were carried out. To determine the normality of the study variables, the Kolmogorov-Smirnov test was performed with the baseline distribution. The use of chi square on the baseline categorical variables is considered a complementary examination to observe if any of the clinical or sociodemographic conditions presented a significantly different distribution between both groups.

  • the second observation have an ethical features. When you identify a patients that, according to this classification, is “not suitable” for surgery ( from a psychological point of view), how  should you use this information? Please, provide some clinical practice use of this informations.

The conclusions paragraph has been modified indicating that the use of the DRAM tool has allowed the development of a liaison program between psychiatry and spinal surgery that has the capacity to offer alternative treatments to surgery for back pain.

I am grateful for your contribution and I hope that this study can improve the multidisciplinary approach to back pain.

Reviewer 2 Report

It has been show in some studies that psychosocial factors are of importance in the screening of patients before spine surgery (Vivian Amaral et al, 2017). It appears obivious that more large-scale research is needed for identifying these factors, as the current state of the literature is limited.

This interesting study evaluates the impact of the DRAM score on the outcome of spine surgery. Several points could be clarified before considering the publication of your manuscript:

  • The statistical methods are not fully described. For example, the use of the Chi2 test to compare the demographic and clinical characteristics between the groups (NDRAM vs DDRAM) but it has not been justified clearly.
  • You showed that there is a significant change in pain scores for the NDRAM group and that the change was not significant for DDRAM group but you did not compare the change between the two groups (i.e. conducting a t-test on the baseline - 6-month variables).
  • To compare two visits, you need to use a paired t-test and not a normal t-test. 
  • Was the normality of distributions verified? If yes, which test was used?
  • The DRAM score is a combination of Zung score and MSPQ score. Why did you not study the predictive value of the Zung score and the MSPQ score separately to see which factor has more impact on the outcome of spine surgery?
  • You conclude in the manuscript that DRAM has "high predictive value". The study only shows that there is a significant association between the DRAM score and spine surgery outcome. Predictive value is measured using specificity and sensitivity or the AUC. Would you please consider including these measures in your paper?

Author Response

To the reviewer number 2:

Thanks you for your comments, I will try to explain the modifications we have done and to answer to your questions:

  • The statistical methods are not fully described. For example, the use of the Chi2 test to compare the demographic and clinical characteristics between the groups (NDRAM vs DDRAM) but it has not been justified clearly.

We have expanded the description of statistical methods to improve the comprehensibility of the operations performed.

  • You showed that there is a significant change in pain scores for the NDRAM group and that the change was not significant for DDRAM group but you did not compare the change between the two groups (i.e. conducting a t-test on the baseline - 6-month variables).

We have expanded the description of statistical methods in this way: “In order to know if both groups had a normal distribution of main study variables (back pain VAS score, leg pain VAS score, SF-12 physical health score and SF-12 mental health score) at baseline we performed a Kolmogorov-Smirnov test. We also performed two paired student’s t-tests (baseline – six weeks and baseline – six months) to compare result variables in the follow-up.”

  • To compare two visits, you need to use a paired t-test and not a normal t-test. 

We used a paired t-test but by mistake in the transcription we did not indicate it in the methods chapter. Thank you very much for pointing this error out to us.Was the normality of distributions verified? If yes, which test was used?

  • The DRAM score is a combination of Zung score and MSPQ score. Why did you not study the predictive value of the Zung score and the MSPQ score separately to see which factor has more impact on the outcome of spine surgery?

The design of our study does not allow us to assess the sensitivity and specificity of the DRAM, it has only allowed us to observe that the patients in the DDRAM group do not present a significant mean improvement after surgery with respect to back pain. In any case, it is a splendid contribution for further study.

  • You conclude in the manuscript that DRAM has "high predictive value". The study only shows that there is a significant association between the DRAM score and spine surgery outcome. Predictive value is measured using specificity and sensitivity or the AUC. Would you please consider including these measures in your paper?

The conclusions paragraph has been modified indicating that the use of the DRAM tool has allowed the development of a liaison program between psychiatry and spinal surgery that has the capacity to offer alternative treatments to surgery for back pain. In this research is not possible to determine sensitivity and specificity parameters

I am grateful for your contribution and I hope that this study can improve the multidisciplinary approach to back pain.
